# Improving Conditional Sequence Generative Adversarial Networks by Stepwise Evaluation

## Abstract

Conditional sequence generation is a widely researched topic. One of the most important tasks is dialogue generation, which is composed of input-output pairs with the one-to-many property. Given the recent success of generative adversarial networks (GANs), GANs have been used for sequence generation. However, there is still limited work of its application on conditional sequence generation. We investigate the influence of GAN on conditional sequence generation with three artificial grammars and dialogue generation. Moreover, we propose stepwise GAN (StepGAN) for conditional sequence generation, which predicts the reward at each time-step. StepGAN can be seen as the general version of SeqGAN. It estimates the expected returns predicted by Monte-Carlo Search in SeqGAN, but it has a lower computational cost than Monte-Carlo Search. Experimental results show that stepwise GAN can outperform other state-of-the-art algorithms in most tasks.

## 1 Introduction

Conditional sequence generation is the task of generating the correspondent response given an input sequence. One of the most important applications is dialogue generation. Dialogue generation is one-to-many; that is, there can be many acceptable responses for a specific input. In previous work, the sequence-to-sequence based dialogue generation model is trained using maximum likelihood estimation, and achieves promising results in terms of both meaning and coherence (Vinyals & Le, 2015). Despite this success, the generated responses given the inputs are still sometimes broken and are often general (for example, *I don't know*). Reinforcement learning was therefore proposed to preserve sequence-level quality as opposed to predicting each word given the sequence history (Ranzato et al., 2015; Kandasamy et al., 2017; Bahdanau et al., 2016).

More recently, generative adversarial networks have been applied to sequence generation, especially for natural language. The discrete nature of random variables for natural language precludes the use of back-propagation. To solve this problem, several approaches have been proposed, such as policy gradient (Yu et al., 2017; Li et al., 2017), Gumbel-Softmax (Kusner & Hernández-Lobato, 2016), MaliGAN (Che et al., 2017), and directly connected WGAN-GP (Gulrajani et al., 2017; Rajeswar et al., 2017; Press et al., 2017). In most previous work, multiple assistant methods are used to stabilize training and often introduce improvements, for example, Monte-Carlo search (Yu et al., 2017; Li et al., 2017; Che et al., 2017) and curriculum learning (Rajeswar et al., 2017; Press et al., 2017). Furthermore, modifications of the GAN objective function have been proposed to improve quality in text generation (Zhang et al., 2017; Lin et al., 2017).

SeqGAN has been successfully applied on dialogue generation (Li et al., 2017). Due to the high variance of SeqGAN with 1-sample estimate REINFORCE algorithm, researchers use Marte-Carlo search for variation reduction. This method costs extremely high computational resources, therefore Reward for Every Generation Step (REGS) is proposed to replace Monte-Carlo search (Li et al., 2017). Nonetheless, REGS results in a less accurate discriminator because it takes non-terminal sequences into consideration.

To address the weaknesses of Monte-Carlo Search and REGS, we propose stepwise GAN (StepGAN). In this approach, the discriminator evaluates the generated sequences at every generation step, and gives a score for every step. A final score for the whole sequence is the summation of the

scores for every time step. This training scheme makes StepGAN a general version of SeqGAN, and can simulate the process of Monte-Carlo search with low extra computational cost. In the proposed approach, both generator and discriminator include weighted factors that change the relative importance of each time step. We find step-time-decreasing weight factors can facilitate the training. This is because the set of hyper-parameters simulate curriculum learning by focusing on generating the head of a sequence. After the first subsequence is fit, further improvement are found in later subsequences.

We construct artificial grammars to assist our realization of GANs in conditional sequence generation. In these tasks, we calculate the accuracy and the coverage of the generated conditioned sequence to evaluate the quality of the model. The coverage is the percentage of the conditioned sequences sampled from the model distribution over all the probable responses. While accuracy reflects coherence and meaningfulness, coverage measures the diversity of the responses. We further compare the proposed approach with several conditional sequence generation approaches on dialogue generation, and evaluate the results by humans. The proposed models are comparable with or even outperform state-of-the-art algorithms.

## 2 RELATED WORK

Conditional sequence generation using the seq2seq model (Sutskever et al., 2014; Vinyals & Le, 2015) has been widely studied, and also for dialogue generation. The model can be learned by maximum-likelihood estimation (MLE), which minimizes the word-level cross-entropy between the true data distribution and the generated approximation. Although this method yields reasonable responses, it suffers from exposure bias and does not take into account sequence-level structure (Ranzato et al., 2015). Exposure bias is introduced because of inconsistent conditions between the training and testing stages: while the ground-truth words are fed to the seq2seq model in the training stage, generated words are used in the testing stage.

To solve these problems with MLE, besides beam search and scheduled sampling (Bengio et al., 2015), (Ranzato et al., 2015) propose the REINFORCE and MIXER algorithms for sequence generation. By providing a task-specific score for the generated sequence, the REINFORCE algorithm (Williams, 1992) guides the seq2seq model to reach higher scores. Because the score is evaluated based on the whole generated sequence, both MLE problems are solved. However, as the REINFORCE algorithm cannot easily train the model from scratch, the MIXER algorithm is proposed to integrate MLE and REINFORCE. In this process, they first train the whole sequence using MLE, after which they accumulate the number of last words trained by REINFORCE. For further improvements, (Bahdanau et al., 2016) adopt another reinforcement learning (Sutton & Barto, 1998) based approach – the actor-critic architecture. They train a critic to predict the expected value of each time step to guide the actor. These algorithms outperform the original MLE algorithm on the task-specific score (BLEU) for text generation. Nonetheless, there is no evidence that these task-specific scores are correlated with human prior knowledge. In particular, the relationship between the scores and human evaluation has been proven weak for dialogue generation (Liu et al., 2016).

Recently, the significant success of generative adversarial networks (GAN) for image processing has led researchers to use GANs for natural language. However, this has seen limited success because of the difficulty of backpropagation through discrete random variables. To address this problem, (Yu et al., 2017) use policy gradients on text generation. The reward is provided by a discriminator with Monte-Carlo search. In addition, (Li et al., 2017) adopt the same idea for dialogue generation. They also propose Reward for Every Generation Step (REGS), which is more time-efficient but is weaker than Monte-Carlo search. Another way to use GAN for natural-language tasks is by using Gumbel-Softmax (Kusner & Hernández-Lobato, 2016), which can simulate the discrete argmax outputs, and be directly backpropagated from the discriminator. Also, MaliGAN (Che et al., 2017) directly derives the gradient estimator for discrete data. More recently, the improved Wasserstein GAN (WGAN-GP) (Gulrajani et al., 2017) has shown success for text generation by directly feeding the softmax layer to the discriminator, even without pre-training. This breakthrough then inspired (Press et al., 2017) and (Rajeswar et al., 2017) to further investigate WGAN-GP for better performance on text generation.

We focus on the influence of different objective function in GANs on conditional sequence generation throughout this paper. We compare the state-of-the-art algorithms without additional assistance

such as teacher forcing, curriculum learning, etc. By this setting, we only consider the improvement attributed by the intrinsic of different algorithms rather than other additional assistances.

# 3 CONDITIONAL SEQUENCE GENERATION

In conditional sequence generation, we generate an output sequence $x \in X$ given an input condition $y \in Y$. When the output sequence is generated from a model, it is denoted $x^G$; when the output sequence is from real data (training examples), it is instead denoted $x^R$. In dialogue generation, both $x$ and $y$ are sequences of words. They can be written as

$$
\begin{aligned}
x &= \{x_t\}_{t=1}^T, x_t \in V \\
y &= \{y_t\}_{t=1}^T, y_t \in V.
\end{aligned}
\tag{1}
$$

Words $x_t$ and $y_t$ represent the word at time step $t$ in the interval $T$ of the specific sequences $x$ and $y$. Set $V$ is the vocabulary set from which the words are selected. In this paper, we generate $x^G$ given $y$ using the seq2seq model as the generator $G$ (Sutskever et al., 2014; Vinyals & Le, 2015). From Sections 3.1 to 3.3, we introduce maximum likelihood estimation, REINFORCE algorithm, and GAN for sequence generation. In Section 4, we introduce the proposed approaches.

## 3.1 MAXIMUM LIKELIHOOD ESTIMATION

The basic idea of maximum likelihood estimation (MLE) for conditional sequence generation is to find the parameters for model $G$ that maximize the likelihood of generating the training data. When using MLE to train the generator model $G$, the objective function is

$$
G^* = \arg\max_G E_{(x^R,y) \sim P^R(X,Y)}[\sum_{t=1}^T \log(P^G(x_t^R|y, x_{1...t-1}^R))],
\tag{2}
$$

where $P^R(X, Y)$ is the joint distribution of the $(x, y)$ pairs in the training data, and $T$ is the length of $x^R$. In the training stage, in Eq. (2), the prediction is learned based on $< y, x_{1...t-1}^R >$, but the condition in the testing stage is $< y, x_{1...t-1}^G >$. This is known as *exposure bias*, which can result in accumulating error when testing. This is also due to the likelihood is estimated at the word level (Ranzato et al., 2015) only as opposed to the whole sequence.

## 3.2 REINFORCE

Conditional sequence generation can be formulated as reinforcement learning. Similar to (Ranzato et al., 2015), we describe it as a Markov decision process (MDP), where state $s$ consists of the condition and previous word sequence – in our case, $s = < y, x_{1...t-1}^G >$ – and an action $a$ is the generated word conditioned on the current state – in our case, $a = x_t^G$ is a word in the vocabulary. Each action is generated according to the policy, which is determined by the parameters of generator model $G$. In typical reinforcement learning, the agent obtains a reward $r_t$ at each time step $t$. In sequence generation, $r_t$ is zero except for $r_T$, which evaluates the goodness of the whole generating $x_{1...T}^G$ given $y$. The generator $G$ learns to maximize the expected reward

$$
G^* = \arg\max_G E_{y \sim P^R(Y), x^G \sim P^G(X|y)}[r_T],
\tag{3}
$$

where $P^R(Y)$ is the probability distribution of condition $y$ in the training data, $P^G(X|y)$ is the probability of generating the sequence $x^G$ given the generator $G$ and condition $y$. Note that the main difference between Equations (2) and (3) is that the condition sequence here is $x_{1...t-1}^G$ rather than $x_{1...t-1}^R$. Moreover, each $x_t^G$ here is sampled using softmax rather than argmax over vocabulary set $V$. The parameters of the generator $\theta_G$ are updated as

$$
\theta_G \leftarrow \theta_G + \eta(r_T - b_t)\nabla \log(p_G(x_t^G|y, x_{1...t-1}^G)),
\tag{4}
$$

where $b_t$ is the baseline to reduce training variance (Sutton & Barto, 1998; Ranzato et al., 2015), and $\eta$ is the learning rate.

### 3.3 GENERATIVE ADVERSARIAL NETWORK

A generative adversarial network (GAN) is composed of a generator and a discriminator (Goodfellow et al., 2014). The discriminator differentiates between real data and data from the generator, and the generator attempts to generate plausible data that will deceive the discriminator. Here we use GAN for conditional sequence generation by considering our model $G$ as the generator and constructing a discriminator $D$ sequentially fed with input-output pairs $y$ and $x$ (Mirza & Osindero, 2014; Li et al., 2017).

#### 3.3.1 SEQGAN

Since the vocabulary $V$ in the generated sequence is a discrete variable, we cannot backpropagate through the generator. In SeqGAN (Yu et al., 2017; Li et al., 2017), the generation task is formulated as a reinforcement learning scenario, similar to that described in Section 3.2, and the reward function is replaced with the discriminator in regular GAN. The discriminator $D$ is then updated through backpropagation, while the generator $G$ is updated using policy gradient with a reward evaluated over $x$ given $y$ by $D$. The optimization functions for $D$ and $G$ are

$$D^* = \arg\max_D E_{y\sim P^R(Y),x^R\sim P^R(X|y)}[\log(D(x^R|y))] + E_{y\sim P^R(Y),x^G\sim P^G(X|y)}[\log(1 - D(x^G|y))]$$

$$G^* = \arg\max_G E_{y\sim P^R(Y),x^G\sim P^G(X|y)}[D(x^G|y)].$$

$$(5)$$

In the basic SeqGAN, $G$ in Equation (5) is optimized using the REINFORCE algorithm. The formulation for optimizing $G$ in Equation (5) is the same as Equation (3), except that $r_T$ is replaced with $D(x^G|y)$.

Due to the sparse reward that only given at the terminal state, this basic setting will cause high training variance. For example, when questioning "What 's your name ?", "I 'm sorry." is a wrong answer, while "I 'm John." is a correct answer. Although they share the same prefix "I 'm", the basic SeqGAN will give the prefix different reward in different sentences. The solution in (Yu et al., 2017; Che et al., 2017) is **Monte Carlo search**. For each prefix $x_{1...t}$, $N$ possible sequences $x_{t+1...T}$ are samples according to the current policy, and the $N$ final rewards are averaged as the reward for current time step. In practice, we have to complete every prefixes for each training data in a batch, and evaluate all of the $mTN$ episodes, where $m$ is the batch size. This method costs extremely high computational resource. For time efficiency, (Li et al., 2017) proposes **Reward for Every Generation Step (REGS)** to replace Monte Carlo search. Because they train the discriminator in REGS with prefixes without considering whether the episode is terminated, REGS causes a less accurate discriminator.

#### 3.3.2 WASSERSTEIN GAN WITH GRADIENT PENALTY (WGAN-GP)

In recent work, WGAP-GP has been successfully used for sequence generation (Gulrajani et al., 2017; Rajeswar et al., 2017; Press et al., 2017). Instead of using more complicate methods such as policy gradient, they directly feed the softmax layer into the discriminator. The generator can therefore be updated through backpropagation. We then formulate the conditional version of WGAN-GP as

$$D^* = \arg\max_D E_{y\sim P^R(y),x^R\sim P^R(X|y)}[D(x^R|y)] - E_{y\sim P^R(y),x^G\sim P^G(X|y)}[D(x^G|y)]$$

$$G^* = \arg\max_G E_{y\sim P^R(Y),x^G\sim P^G(X|y)}[D(x^G|y)].$$

$$(6)$$

## 4 PROPOSED APPROACH: STEPWISE GAN

The basic idea of stepwise GAN (see Fig. 1), or StepGAN, is to construct a sequence-to-sequence model as discriminator $D$. At each time step of $D$'s decoder, the hidden vector is passed to a fully-connected layer. The discriminator $D$ then outputs the evaluation score for each subsequence $\langle y, x_{1...t}\rangle$, denoted as $D(x_{1...t}|y)$. With discriminator $D$, we seek to minimize the summation of $D(x^G_{1...t}|y)$ over the time steps when input the generated sequences $x^G$. Simultaneously, $D$ maximizes the summation of $D(x^R_{1...t}|y)$ when the input is real data $x^R$. The optimization of $D$ and $G$ is

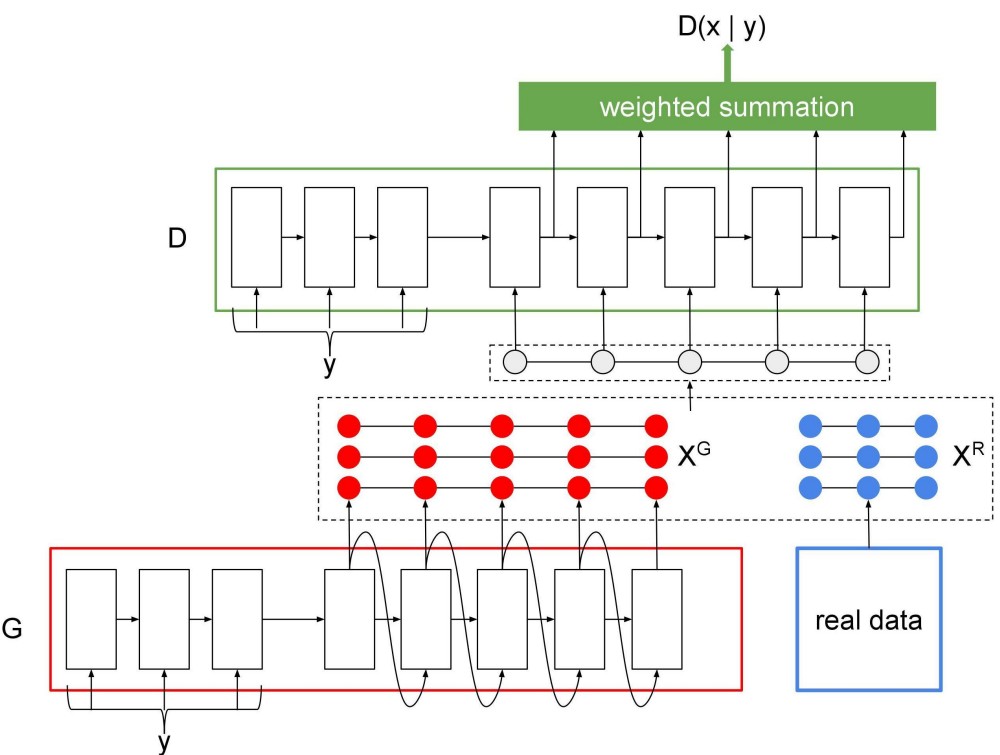

Figure 1: Illustration of stepwise GAN

thus

$$D'(x|y) = \sum_{t=1}^{T} \alpha_t^D D(x_{1...t}|y)$$

$$D^* = \arg\max_D E_{y \sim P^R(y), x^R \sim P^R(X|y)}[\log(D'(x^R|y))]$$

$$+ E_{y \sim P^R(y), x^G \sim P^G(X|y)}[\log(1 - D'(x^G|y))], \quad (7)$$

$$G^* = \arg\max_G E_{y \sim P^R(Y), x^G \sim P^G(X|y)}[D(x^G|y)],$$

where $\alpha_t^D$ is a weighted factor with $\sum_{t=1}^{T} \alpha_t^D = 1$. In practice, we set $\alpha_t^D = \frac{1}{T}$, but it is possible to give different steps different weights. StepGAN not only considers terminated episodes but also assign scores for each prefix. Although we only use one way to train $D$, we think there are two ways to interpret the $D$'s scores, and different view points lead to different update formulations for $G$:

- $D(x_{1...t}|y)$ evaluates the extra benefit of adding the word $x_t$ into the sequence. The formulation for $G$'s parameter update is

$$\theta_G \leftarrow \theta_G + \eta\alpha_t^G (\sum_{t'=t}^{T} D(x_{1...t'}|y))\nabla\log(p_G(x_t^G|y, x_{1...t-1}^G)). \quad (8)$$

  $G$ has to increase the summation $\sum_{t'=t}^{T} D(x_{1...t'}|y)$. We call this stepGAN-Seq.

- $D(x_{1...t}|y)$ evaluates the average goodness of all the sequences beginning with $x_{1...t}$. The update formulation for $\theta_G$ is

$$\theta_G \leftarrow \theta_G + \eta\alpha_t^G D(x_{1...t}|y)\nabla\log(p_G(x_t^G|y, x_{1...t-1}^G)). \quad (9)$$

  The $G$ only needs to learn to increase $D(x_{1...t}|y)$. We call this stepGAN in following.

Using factor $\alpha_t^G$, we diversify training by arbitrarily weighting the importance of each time step[1]. We explore the influence of different values of $\alpha_t^G$ and compare the two update formulations in section 5 and appendix B.

StepGAN-Seq is a generalized version of SeqGAN. If we set $\alpha_T^D = 1$, $\alpha_t^D = 0$ for $t < T$, and $\alpha_t^G = 1$ for all $t$, then stepGAN-Seq is equivalent to SeqGAN without Monte Carlo search. Also, REGS (Li et al., 2017) can be induced by set $\alpha_t^D = 1$ at a randomly chosen time step.

StepGAN is similar to the actor-critic architecture in (Bahdanau et al., 2016). Instead of the assigned task-specific score, we learn the score by adversarial learning. Because $D$'s scores are the expected return in this setting, we would like StepGAN to approximate the expected return obtained by Seq-GAN or MaliGAN with Monte Carlo search. This approach only need to add a set of weight factors, and therefore much time efficient than Monte Carlo search. For more details of the algorithm, please refer to appendix A for pseudo-code.

## 5 Experiments

We use a recurrent neural network for both the discriminator and generator due to its strong sequential correlation (Press et al., 2017; Rajeswar et al., 2017). Specifically, we use gated recurrent units (GRUs) (Chung et al., 2014) in our experiments. We view the noise feature in GAN as the random process of sampling from the softmax layer distribution.

### 5.1 Artificial Grammars

To better evaluate GANs for conditional sequence generation, we define three artificial grammars: sequence, counting, and addition. The three grammars are described in Table 1. For the sequence grammar, the aim is to generate a continued consecutive number sequence behind the input $Y$. For example, for input $\langle 1, 2, 3 \rangle$, the answer would be a consecutive number sequence of any length starting with 4, such as $\langle 4, 5, 6, 7, 8 \rangle$. The counting grammar is more complicated. The generated sequence should contain exactly 3 words, where the median is a randomly selected word from the input sequence. The first generated word should be the number of words on the left-hand side of the selected median, while the last generated word should be the number of words on the right-hand side. For example, when the input is $\langle 5, 9, 2, 8, 3, 2, 9, 1 \rangle$, one permissible generated sequence is $\langle 0, 5, 7 \rangle$. Last, for the addition grammar we generate the addition of two numbers randomly segmented from the input sequence. That is, for input $\langle 8, 1, 3, 4 \rangle$, then one permissible output is the addition of 8 and 134 – thus $\langle 1, 4, 2 \rangle$. Note that both the input and output numbers for this grammar are represented in terms of their corresponding digits.

The purpose of these design is to imitate major properties in dialogue generation, such as variable-length, repeated prefixes, the same sequence space, one-to-many, and many-to-one. The variable-length property means there is no fixed length for the input and output sequences. The repeated prefixes property means the beginning subsequences are usually shared by many data, for instance *What* and *I am* in natural language. The same sequence space here means that the input and output have the same structure and as such are sampled from the same space. Finally, one-to-many and many-to-one are quite common in dialogue generation. For example, when asking *How are you?*, responses vary from *I'm fine* to *Great! How are you?*. Also, the same response can be paired with multiple questions, such as for *My name is Paul* in response to *What's your name?* and *Who are you?*.

We then randomly generate 100,000 samples as training data, 10,000 as development data, and 10,000 samples as testing data. The architectures are set to one layer with 128 hidden units. We evaluate our results using the three measures in Table 2. The first is the accuracy of samples generated from the argmax policy (Acc), the second is that generated from the softmax probability (AccS), and the last is the coverage of softmax samples over all the permissible answers of the specific grammar (Cov). We report them to ensure whether the one-to-many property is being learned. AccS and Cov are important because they can indicate if the model can learn the underlying distribution of answers. When mode collapse happens, which means the model only know a specific

---

[1] $\alpha_t^G$ needs not be equivalent to $\alpha_t^D$.

Table 1: Grammar definitions and examples

| Grammar | Definition | Examples |
|---|---|---|
| Sequence | Continue the sequence for a random length | 123: 4, 45, ... |
| Counting | Randomly choose a digit, and then calculate the left- and right-hand lengths | 123: 012, 121, 230 |
| Addition | Randomly partition, and then add the two numbers | 123: 15, 24 |

Table 2: Results of artificial grammars with different algorithms. Evaluation label Acc (%) is the accuracy of argmax samples, AccS (%) is the accuracy of softmax samples, and Cov (%) is the coverage of softmax samples over permissible answers. The dash (-) here indicates that the algorithm introduced no improvements based on the pre-trained model.

| | Sequence | | | Counting | | | Addition | | |
|---|---|---|---|---|---|---|---|---|---|
| | Acc | AccS | Cov | Acc | AccS | Cov | Acc | AccS | Cov |
| MLE | 97.43 | 81.32 | 53.77 | 73.48 | 68.89 | 70.63 | 44.57 | 32.28 | 31.79 |
| *REINFORCE* | *99.81* | *97.30* | *4.54* | *99.97* | *99.36* | *16.99* | *79.98* | *75.60* | *18.32* |
| WGAN-GP | - | - | - | - | - | - | - | - | - |
| basic-MaliGAN | 97.34 | 81.66 | 54.19 | 74.12 | 70.35 | 70.15 | 44.27 | 32.05 | 31.92 |
| basic-SeqGAN | 97.20 | 80.28 | 57.61 | 74.59 | 70.56 | 70.39 | 44.87 | 32.30 | 31.90 |
| MC-SeqGAN | 97.20 | 80.98 | 55.49 | 72.96 | 68.10 | 70.54 | 44.72 | 32.28 | 31.83 |
| REGS | 97.42 | 81.36 | 53.82 | 75.99 | 70.93 | 69.40 | 44.64 | 32.32 | 32.01 |
| StepGAN-Seq | 97.11 | 74.85 | **67.49** | 75.47 | 70.64 | 69.82 | **45.55** | 32.49 | 32.01 |
| StepGAN | 97.19 | 75.92 | **66.08** | **81.98** | 72.24 | 69.02 | **44.94** | 32.19 | 31.67 |

type of answers, it will obtain high AccS and low Cov scores. To ensure a fair comparison, all the algorithms are based on the same pre-trained model: the MLE model listed in Table 2.

We discuss the results of MLE, REINFORCE, and state-of-the-art GAN algorithms on Sequence, Counting, and Addition in Table 2. REINFORCE has higher Acc and AccS than MLE[2], but it results in strong mode-collapse (very low Cov). We have a very strong MLE baseline for Sequence. Therefore we cannot pretrain discriminator well based on this baseline MLE model. Every GAN algorithms cannot outperform MLE by this setting. StepGAN improves Acc on both Counting and Addition without heavily trade-off with Cov. That is, training model using StepGAN enhances and maintains the knowledge of underlying distribution rather than resulting in strong mode-collapse as REINFORCE.

To investigate the trade-off between AccS and Cov, we plot the accuracy-coverage curve in Fig. 2. The trade-off between accuracy and coverage is controlled by sharpening the softmax layer. Besides Fig. 2a, of which the GANs do not obtain good results, StepGAN improves the accuracy-coverage curves in Fig. 2b and Fig. 2c. This is consistent with our realization of Table 2 that StepGAN fits model to the underlying distribution better.

## 5.2 DIALOGUE GENERATION

We split OpenSubtitles (Tiedemann, 2009) into training set, development set, and testing set with a vocabulary of the top 4,000 most frequently occurring words. Both the generator and discriminator are 1-layer GRUs with a hidden dimension set to 512[3]. To compare the improvements introduced by all the algorithms, we first pre-trained the generator using MLE, after which we further trained the model for 1-epoch by different GANs. SeqGAN, MaliGAN, REGS, and StepGAN were compared. We do not compare Monte Carlo search with other approaches because its time complexity is much larger. All discriminators of SeqGAN, MaliGAN, REGS, and StepGAN were pre-trained on real

---

[2]Since the given reward of REINFORCE is the true accuracy, the Acc and AccS of REINFORCE here are taken as the upper bounds.

[3]We trained another value network with the same seq2seq architecture to estimate the baseline. This is similar to (Li et al., 2017).

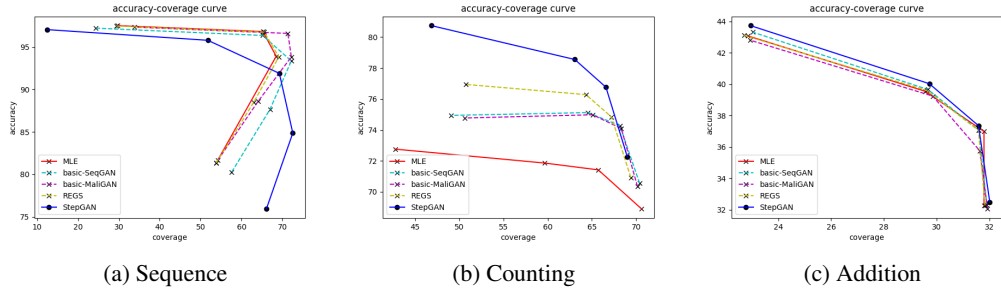

| (a) Sequence | (b) Counting | (c) Addition |

Figure 2: Sampled accuracy and coverage curves

Table 3: Human evaluation and BLEU score for dialogue generation. CoHS (%) is coherence human score. SHS (%) is sentence structure human score.

|  | CoHS (%) | | | SHS (%) | | | BLEU | | |
|---|---|---|---|---|---|---|---|---|---|
|  | Argmax | BS | MMI | Argmax | BS | MMI | Argmax | BS | MMI |
| **MLE** | 44.89 | 54.22 | **60.44** | 15.11 | 1.33 | 7.56 | 0.222 | 0.281 | 0.272 |
| **SeqGAN** | 41.33 | 53.33 | **63.55** | 30.67 | 6.22 | 10.22 | 0.202 | 0.267 | 0.251 |
| **MaliGAN** | 35.56 | 51.11 | 45.33 | 20.89 | 5.78 | 8.00 | 0.180 | 0.271 | 0.263 |
| **REGS** | 36.44 | 54.67 | 53.78 | 36.44 | 9.33 | 9.78 | 0.180 | 0.256 | 0.246 |
| **StepGAN** | **47.56** | **63.56** | 61.33 | **40.89** | 3.56 | 8.89 | 0.171 | 0.254 | 0.248 |

data and generated data from the pre-trained generator. Note that these models were all trained without MIXER, curriculum learning, or teacher forcing, etc. Both the generator and discriminator are optimized by SGD. We used grid search in the experiments with learning rate={1e-1,1e-2,1e-3}, discriminator iteration step={1,5}, and used batchsize=64.

For human evaluation, we randomly selected 25 inputs from the testing set, and decoded using argmax policy, beam search, and MMI (Li et al., 2015)[4]. We presented both an input and the generated outputs to 8 and 4 judges respectively, and we asked them to do Turing test (correct or not) of the coherence and sentence structure. Coherence is the rationality of the generated responses given inputs. Sentence structure is the correctness and complexity of grammar. The sentences provide specific information would be considered as having better sentence structure rather than the general ones (egs. I don't know.) due to more complex grammar. In Table 3, the two measures are labeled as CoHS (Coherence Human Score) and SHS (Sentence structure Human Score). We also show the BLEU score of each algorithm. It was already found that BLEU score is inconsistent with human evaluation (Liu et al., 2016), we also observe the same phenomenon it in our experiments.

We show the CoHS and SHS of 15 different results (5 different algorithms and 3 different decoding methods) in Table 3. First, we can see decoding using beam search or MMI improve CoHS. StepGAN obtains the best performance in terms of CoHS when using argmax policy or beam search, but StepGAN cannot further increase the performance using MMI. When using MMI, the CoHS of MLE, SeqGAN and StepGAN are comparable. Second, argmax policy has higher SHS than beam search and MMI in all cases. In the meantime, GANs have higher SHS than MLE, and StepGAN has the highest score with argmax. The inconsistency of improvement between argmax policy, beam search and MMI is very likely because that the GANs are learned with the softmax policy and do not consider beam search and MMI during training. Additionally, we know beam search and MMI maximize the probability of response given an input without maintaining the probability of the response itself. This makes them prefer a coherence response rather than a good sentence structure. These statistics show that SeqGAN, MaliGAN, and REGS cannot consistently improve both coherence and sentence structure, whereas StepGAN outperforms MLE in terms both CoHS and SHS with all decoding methods.[5]

---

[4]We use MMI-p(x) in our experiments.
[5]To see the generated examples, please refer to appendix D.

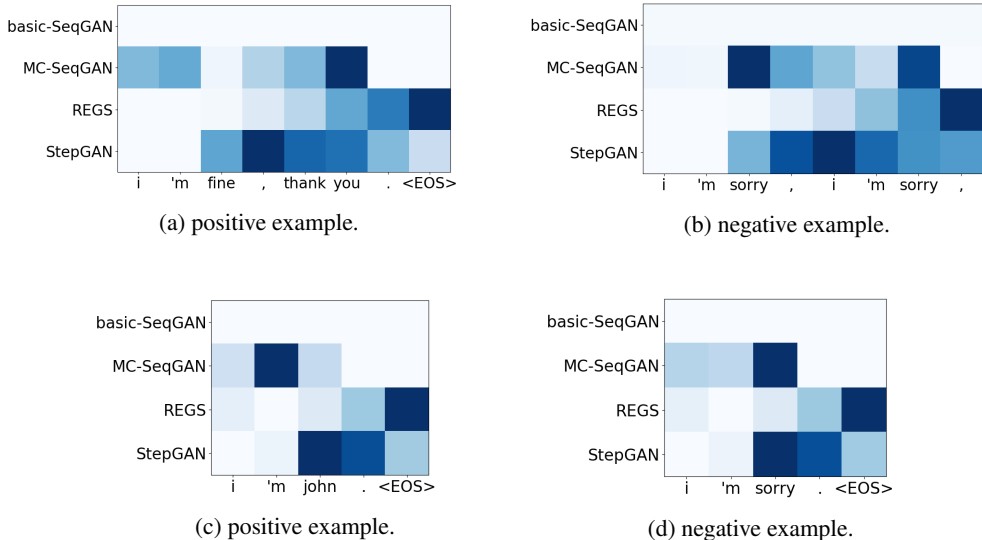

(a) positive example.

(b) negative example.

(c) positive example.

(d) negative example.

Figure 3: The variation of discriminators' scores using different GAN algorithms during training iterations. The printed color is normalized throughout the generation steps (x-axis) for each algorithm. (a)(b) are given "how are you ?" as input, and (c)(d) are given "what 's your name ?" as input.

To understand what the discriminators learn, we measure the variance throughout the training iterations at each generation step. In adversarial learning, discriminator's scores oscillate during training according to the current performance of generator. We argue that discriminator's score for the most crucial generation step is the most easy to oscillate. This is because the generation step is the most important one for discriminator to identify whether it's real or fake. In Fig. (3), we show four examples. Fig. (3a) and Fig. (3b) are respectively true response and wrong response given input question *"how are you ?"*, and Fig. (3c) and Fig. (3d) are given input question *"what 's your name ?*. The darker color indicates the higher variance on the generation step.

In Fig. (3), the colors for SeqGAN is always the same. Because the disriminator of SeqGAN only evaluate the whole sequence, the generation steps means no difference to discriminator. Second, REGS and StepGAN both aim to approximate Monte-Carlo search on SeqGAN, but in practice, we can clearly see that the variance of REGS is very different from Monte Carlo search. This is because REGS considers non-terminated episodes, which makes REGS has to spend extra effort on the generation maximum length to check whether there's a terminal state ($< EOS >$). The results of StepGAN and Monte Carlo search (MC-SeqGAN) are quite similar. Based on MC-SeqGAN and StepGAN, the important parts (with darker colors) in the sentences for discriminating the true ones from fake correspond to human knowledge. For example, *"fine , thank you "* are the most important region in Fig. (3a), *"sorry , i 'm sorry"* are the most important region in Fig. (3b) when answering *"how are you ?"*. When given *"what 's your name ?"*, MC-SeqGAN and StepGAN focus on *"'m john"* in Fig. (3c) and *"sorry"* in Fig. (3d). We believe the success of StepGAN comes from estimating the goodness of a sequence at every generation step as Monte Carlo search, but with little extra computation.

# 6 CONCLUSION

In this paper we propose StepGAN to approximate Monte Carlo search with a much lower computational cost.We show that the proposed StepGAN performs equally to or outperforms the state-of-the-art GAN algorithms on artificial grammars. On a representative real-world conditional sequence generation task–dialogue generation, StepGAN also outperforms other approaches on both coherence and sentence structure.

Our proposed artificial grammars not only accurately reflect model coverage and accuracy but also boast clearly distinguishable styles. For example, the sequence style can be the length, the counting style can be the selected digit position, and the addition style can be the selected partition position. This property lends itself to investigating style transfering for sequences generation, which is one of our aims for future work.

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

## A  APPENDIX: PSEUDO CODE OF STEPWISE GAN

Table 4: Pseudo code of stepwise GAN

**Algorithm 1** Stepwise GAN (StepGAN) Training

| | |
|---|---|
| 1 | **for** number of training iterations **do** |
| 2 |     **for** i=1, D-steps **do** |
| 3 |         Sample $(y, x^R)$ from real data |
| 4 |         Sample $x^G \sim P^G(.|y)$ |
| 5 |         Update D using equation (7) |
| 6 |         $D'(x|y) = \sum_{t=1}^{T} \alpha_t^D D(x_{1...t}|y)$ |
| 7 |         $D^* = \arg\max_D E_{y\sim P^R(y), x^R\sim P^R(X|y)}[\log(D'(x^R|y))]$ |
| |            $+E_{y\sim P^R(y), x^G\sim P^G(X|y)}[\log(1 - D'(x^G|y))]$ |
| 8 |     **end for** |
| 9 |     **for** i=1, G-steps **do** |
| 10 |         Sample $y$ from real data |
| 11 |         Sample $x^G \sim P^G(.|y)$ |
| 12 |         Update G using equation (9) |
| 13 |         $\theta_G \leftarrow \theta_G + \eta\alpha_t^G D(x_{1...t}|y)\nabla log(p_G(x_t^G|y, x_{1...t-1}^G))$ |
| 14 |     **end for** |
| 15 | **end for** |

## B  WEIGHTED FACTORS SEARCH OF STEPGAN AND STEPGAN-SEQ

Table 5: StepGAN and StepGAN-Seq with different weight factors. The dash (-) here notes that the weight factors yield neither improvements nor deterioration.

| | | StepGAN | | | StepGAN-Seq | | |
|---|---|---|---|---|---|---|---|
| | | **Acc** | **AccS** | **Cov** | **Acc** | **AccS** | **Cov** |
| **Sequence** | Uniform | 97.19 | 75.92 | 66.08 | 97.11 | 74.85 | 67.49 |
| | Increase | 97.17 | 75.33 | 66.75 | 97.18 | 73.30 | 68.19 |
| | Decrease | 97.16 | 74.73 | 67.14 | 97.09 | 74.34 | 67.48 |
| **Counting** | Uniform | 77.23 | 71.43 | 70.18 | 74.91 | 70.70 | 69.84 |
| | Increasing | 74.50 | 69.66 | 70.86 | 74.54 | 70.74 | 70.11 |
| | Decreasing | **81.98** | 72.24 | 69.02 | **75.47** | 70.64 | 69.82 |
| **Addition** | Uniform | 44.26 | 32.39 | 31.91 | 44.77 | 32.34 | 31.65 |
| | Increasing | - | - | - | - | - | - |
| | Decreasing | **44.94** | 32.19 | 31.67 | **45.55** | 32.49 | 32.01 |

We compare different weighted factors $\alpha^G$ for StepGAN and StepGAN-Seq, with $\alpha_t^D = \frac{1}{T}$ for all the presented cases. As depicted in Table 5, the three grammars are trained using three sorts of weighted factors: uniform ($\alpha_t^G = 1$), increasing ($\alpha_t^G = t$), and decreasing ($\alpha_t^G = T - t + 1$). The results clearly show that the time-step-decreasing weight factors positively affect training. We believe this is because the training spirit of decreased weighted factor start from first correcting prefix. After correcting prefix, it becomes easier to correct the suffix. Please refer to section 5 if you are interested in the details of Acc, AccS, and Cov.

## C  ENERGY-BASED STEPWISE GAN (EBSTEPGAN)

We propose energy-based stepwise GAN (EBStepGAN) that only change the form of objective function of StepGAN. This is mainly inspired by energy-based GAN (Zhao et al., 2016). As depicted in Fig. 4, the discriminator $D$ here has the same architecture as generator $G$, and its energy function is cross-entropy. Discriminator assigns low energy to real samples and high enery to generated

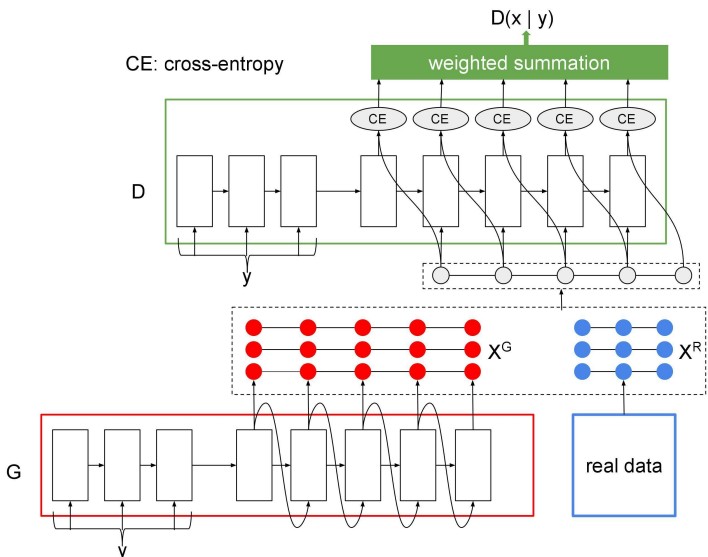

Figure 4: Illustration of energy-based stepwise GAN

samples. The advantage of EBStepGAN is that we can initialize both generator and discriminator with the same pre-trained model using MLE. The optimization functions are written as

$$D^* = \arg\max_D E_{(x^R,y)\sim P^R(X,Y)}[\sum_{t=1}^{T} log(P^D(x_t^R|y, x_{1...t-1}^R))]$$

$$+ maximum(0, \beta - E_{y\sim P^R(Y), x^G\sim P^G(X|y)}[\sum_{t=1}^{T} log(P^D(x_t^G|y, x_{1...t-1}^G))])$$

(10)

$$G^* = \arg\max_G E_{y\sim P^R(Y), x^G\sim P^G(X|y)}[\sum_{t=1}^{T} log(P^D(x_t^G|y, x_{1...t-1}^G))],$$

(11)

where $P^D(x_t^G|y, x_{1...t-1}^G)$ in Equation (11) is the probability of generating $x_t^G$ given $< y, x_{1...t-1}^G >$ based on the current model $D$, and $\beta$ in Equation (10) is the threshold for preventing the discriminator from distinguishing generated samples from real samples too easily. When the energy of generated samples is too high, the threshold turns off the second term in (10). The results of EBStepGAN is shown in Table 6. According to Table 6, EBStepGAN does not show better performance than StepGAN in our current experiments.

Table 6: Results of artificial grammars by EBStepGAN. Please check Table 2 for other results.

|  | Sequence | | | Counting | | | Addition | | |
|---|---|---|---|---|---|---|---|---|---|
|  | Acc | AccS | Cov | Acc | AccS | Cov | Acc | AccS | Cov |
| EBStepGAN | 97.32 | 80.09 | 57.12 | 79.78 | 71.52 | 68.44 | 45.74 | 32.68 | 32.11 |

# D APPENDIX: EXAMPLES OF NEURAL DIALOGUE GENERATION

Table 7: Examples of neural dialogue generation trained on OpenSubtitles

| | |
|---:|:---|
| **Input** | hello , i 'm senator snatch . |
| **MLE** | i 'm sorry . |
| **SeqGAN** | i 'm sorry , i 'm sorry . |
| **MaliGAN** | i 'm a good man . |
| **REGS** | i 'm here to see you again |
| **StepGAN** | sorry you 're late . |
| **Input** | i got the money right here . |
| **MLE** | i 'm sorry . |
| **SeqGAN** | i 'm here to help . |
| **MaliGAN** | i can . |
| **REGS** | i can 't do it . |
| **StepGAN** | i 'll take you home . |
| **Input** | hope you enjoyed your pee ... because it 's your last on this trip . |
| **MLE** | it 's not a problem . |
| **SeqGAN** | it 's not your fault . |
| **MaliGAN** | it doesn 't matter . |
| **REGS** | it 's not a dream . |
| **StepGAN** | it 's the only thing i can do . |
| **Input** | he say ... if you want to be an actor , he has something to show you . |
| **MLE** | you 're not . |
| **SeqGAN** | you can 't do that . |
| **MaliGAN** | what 's he mean ? |
| **REGS** | he 's a very good guy , you know . |
| **StepGAN** | you 're not gonna be a liar . |
| **Input** | thank you , thank you . |
| **MLE** | i 'm sorry . |
| **SeqGAN** | i 'm sorry , sir . |
| **MaliGAN** | thank you . |
| **REGS** | i 'll be right back . |
| **StepGAN** | you 're welcome . |
| **Input** | i feel like a paris princess . |
| **MLE** | i know it . |
| **SeqGAN** | i know that you 're not there . |
| **MaliGAN** | i feel like ... |
| **REGS** | i can 't see it . |
| **StepGAN** | you 're not a child anymore . |
| **Input** | i 'm tired of all your questions . |
| **MLE** | i 'm gonna have to ask you to come . |
| **SeqGAN** | i 'm gonna call you . |
| **MaliGAN** | i 'm a friend . |
| **REGS** | i 'm going to have to ask you to come in |
| **StepGAN** | i 'm going to be a friend of you . |
| **Input** | it 's great . |
| **MLE** | yeah . |
| **SeqGAN** | it 's great , huh ? |
| **MaliGAN** | it 's great . |
| **REGS** | i was just thinking about it . |
| **StepGAN** | you 're so sweet . |

