# OpenReview forum: "Improving Conditional Sequence Generative Adversarial Networks by Stepwise Evaluation"
_ICLR.cc/2018/Conference — Reject_

### Official Review · AnonReviewer3 · 2017-11-26
**The paper is concerned with improving the sequence generation problem, in particular, for dialogue generation problem. The main contribution is in proposing to compute the cumulative reward for each step in the sequence generation procedure. The paper demonstrates that this leads to better performance in artificially generated grammar and for dialogue generation.**

**Rating:** 4
**Confidence:** 3

**Review:**

Quality: The paper proposes a direct improvement over SeqGAN by Yu. et. al. (2017). My assessment is partially determined by comparing this paper to Yu et. al (2017). In my opinion, this paper is lacking in quality in comparison to Yu et. al (2017). In particular, Yu et. al. (2017) provides detailed derivation of the policy gradient accompanied by a pseudo-code (algorithm) on how one can implement SeqGAN. On the contrary, this paper does not provide such details. Perhaps, all of the details of SeqGAN follows immediately, but the paper should not assume that all readers will be familiar with SeqGAN.

Clarity:

1. The paper provides a review of related methods on conditional sequence generation in Section 3. However, it is very brief and as a non-expert in this field, I needed to refer to the original papers anyways. Perhaps, the review of the related methods can go to the Appendix and this space can be better utilized to expand on the original contributions made by the paper.
2. MCMC (Markov chain Monte Carlo) is mentioned in 4.1 but it is not explained.
3. Figure 1 is not sufficiently explained; neither in text nor in the figure caption. It would help greatly to describe the details of the network architecture shown in this figure.

Originality: The paper proposes a generalization of SeqGAN; however, in my opinion, the methodological contribution appears to be only incremental on SeqGAN.

Significance: The paper's significance may be evaluated in terms of its impact on applications as it proposes an improvement over the previous work of SeqGAN. However, the extent to which the evaluation is carried out is somewhat unsatisfactory with only one real application. Also, the applications considered in the experiments are primarily on dialogue generation. My initial impression is that the methodology lacks generality and may perhaps cater better to domain specific publication venues.

---

> ### Author Response · Authors · 2018-01-05
> **Response to Reviewer3**
>
> Quality: The paper proposes a direct improvement over SeqGAN by Yu. et. al. (2017). My assessment is partially determined by comparing this paper to Yu et. al (2017). In my opinion, this paper is lacking in quality in comparison to Yu et. al (2017). In particular, Yu et. al. (2017) provides detailed derivation of the policy gradient accompanied by a pseudo-code (algorithm) on how one can implement SeqGAN. On the contrary, this paper does not provide such details. Perhaps, all of the details of SeqGAN follows immediately, but the paper should not assume that all readers will be familiar with SeqGAN.
>
>
> Ans: Thanks for your advice. We have add pseudo-code in Appendix A.
>
> Clarity:
>
> 1. The paper provides a review of related methods on conditional sequence generation in Section 3. However, it is very brief and as a non-expert in this field, I needed to refer to the original papers anyways. Perhaps, the review of the related methods can go to the Appendix and this space can be better utilized to expand on the original contributions made by the paper.
> Ans: Thanks for your suggestion. We have rewrote Section 3 for better explaination.
>
> 2. MCMC (Markov chain Monte Carlo) is mentioned in 4.1 but it is not explained.
> Ans: We have added the explaination in Section 3.3.1: SeqGAN.
>
> 3. Figure 1 is not sufficiently explained; neither in text nor in the figure caption. It would help greatly to describe the details of the network architecture shown in this figure.
> Ans: We have rewrote Section 4. Hope that is more clear.
>
> Originality: The paper proposes a generalization of SeqGAN; however, in my opinion, the methodological contribution appears to be only incremental on SeqGAN.
>
> Ans: We think the main contribution and originality of this paper are proposing a method to approximate Monte Carlo search with little computational cost.
>
> Significance: The paper's significance may be evaluated in terms of its impact on applications as it proposes an improvement over the previous work of SeqGAN. However, the extent to which the evaluation is carried out is somewhat unsatisfactory with only one real application. Also, the applications considered in the experiments are primarily on dialogue generation. My initial impression is that the methodology lacks generality and may perhaps cater better to domain specific publication venues.
>
> Ans: Thanks for your indication. While we do not have much time to conduct many different real-world tasks, we think approximating Monte Carlo search might be another significance. There are some other points we want to clarify. First, we think that this paper serves more as an evaluation on conditional sequence generation rather than the original work of SeqGAN, which is sequence generation without condition. Second, we mainly apply to dialogue generation because it’s one of the typical 1-to-many conditional sequence generation task and one of our most important baseline “REGS” is proposed on this.

---

### Official Review · AnonReviewer1 · 2017-11-27
**Interesting and potentially powerful approach to language generation, but incomplete and lacking insight / more thorough evaluation**

**Rating:** 5
**Confidence:** 4

**Review:**

UPDATE, 1/11/18:
I read the revision. The writing has improved, and the new experiments are good. That said, it is my opinion that the paper is still not quite ready for publication, mostly because the story behind the model doesn't lead to the conclusions being made from the experiments in a clean and consistent way. It's a bit like patchwork at this point. The authors I think will need to put some time into a rewrite, but the content itself is worth pushing forward.

Some notes:
"StepGAN a general version of SeqGAN, and can simulate the process of Monte-Carlo search with low extra computational cost:" This really is a strong claim that's not proven in any way in the paper.
"In typical reinforcement learning, the agent obtains a reward...": in typical RL settings, it's just as likely to find single or episodal rewards and this setting isn't limited to those where you have an extrinsic reward at each step.
Why do you still have WGAN-GP when there are no accompanying numbers?

/begin old review
The approach is interesting, but as a contribution the paper has a long way to go. The ideas are there and everything seems correct, but there’s little motivation / insight on the model and why it might be better than competing methods for NLP tasks.

It would be good to see some sort of concrete analysis as far as what the model is doing (for instance how the discriminator scores change), and a comparison of how the reward signal given here might differ from other methods (SeqGAN, MaliGAN), and why this might be better. All we get is some scores, but it’s never clear why these scores indicate good (conditional) language generation. Can we not also look at BLEU scores for language generation or some other metric?

Finally, the writing need to be improved: it starts out OK, but it progressively gets worse and worse.

Detailed notes:
P2
It might be good to mention beam search and scheduled sampling as other common methods to address the exposure bias.
“objective function irrelevant to backpropagation”: what does this mean?
MaliGAN actually also uses a “policy gradient”, which corresponds to an estimate of the likelihood ratio, to address the discrete problem.
Though distinct from this work, Gulrajani used a CNN.

P3
Use \log for logarithm
“Moreover, the likelihood is only estimated at word-level”: is this true? It seems to me that likelihood of the sequence is estimated as well.

P5
Why is this a generalized version of SeqGAN? The claim the discriminator value D(x_1..t | y) is the same as what you would get from a full-sequence generator using MCMC seems like a stretch.
Did you not use a baseline?
You might want to build in a little more motivation for these different update rules. I think I understand that (10) is meant to accumulate credit across the rest of the sequence, while (11) does not, but it would be good to have this clearly stated. Why do you think one would work better than the other?

P6
“The generator G, in the mean time, struggle to maximize the likelihood of discriminator D” I don’t understand what this means.
What was the motivation for using the same model here? Is the energy in this formulation related in any ways to EBGAN? Could you do something similar with separate parameters? Why would be or why would this not be a good idea?
I do like these synthetic tasks. I think that more analyses would be helpful in understanding what the model (and what competing models) are doing.

P7:
What is VLGAN in the table?
Perhaps it would be worth exploring changing alpha through optimization?

P8:
It seems like many of the improvements in the table are marginal (with some exceptions): is it possible that ESGAN was optimized better?
“auxiliary tricks” I would avoid this wording.

Other comments on experiments:
It seems like the actual NLP part of this paper is quite sparse. Why was MaliGAN left out of the real experiments?

---

> ### Author Response · Authors · 2018-01-05
> **Response to Reviewer1 - part 1**
>
> The approach is interesting, but as a contribution the paper has a long way to go. The ideas are there and everything seems correct, but there’s little motivation / insight on the model and why it might be better than competing methods for NLP tasks.
> It would be good to see some sort of concrete analysis as far as what the model is doing (for instance how the discriminator scores change), and a comparison of how the reward signal given here might differ from other methods (SeqGAN, MaliGAN), and why this might be better. All we get is some scores, but it’s never clear why these scores indicate good (conditional) language generation. Can we not also look at BLEU scores for language generation or some other metric?
>
> Ans: Thanks for your suggestion. We propose to use the variation throughout the training process of discriminator to check which part is the most crucial one for the method (SeqGAN, Monte Carlo search, REGS, StepGAN). The reason we didn’t choose MaliGAN because its objective function is different from SeqGAN, from which REGS and StepGAN derived. We found that StepGAN can better approximate the crucial part as Monte Carlo search than REGS, and the estimated crucial parts are also correspondent to human knowledge. The details are described in Section 5.2.
>
> Finally, the writing need to be improved: it starts out OK, but it progressively gets worse and worse.
>
> Ans: Because none of the authors are English native speakers, we hire an English speaker with computer science Ph.D to polish the English writing.
>
> Detailed notes:
> P2
> It might be good to mention beam search and scheduled sampling as other common methods to address the exposure bias.
>
> Ans: We will cite the related paper.
>
> “objective function irrelevant to backpropagation”: what does this mean?
> MaliGAN actually also uses a “policy gradient”, which corresponds to an estimate of the likelihood ratio, to address the discrete problem.
> Though distinct from this work, Gulrajani used a CNN.
>
> Ans: Thanks for your indication. I thought MaliGAN directly derived the gradient estimator, which has the same form as policy gradient, instead of evaluated a reward for policy gradient. This might be my uncarefulness in explanation.
>
> P3
> Use \log for logarithm
>
> Ans: Thanks for your suggestion. We have revised this.
>
> “Moreover, the likelihood is only estimated at word-level”: is this true? It seems to me that likelihood of the sequence is estimated as well.
>
> Ans: This is my interpretation. The likelihood is estimated over the whole sequence, however, the minimization is conducted on word-level when training. This is like the description of training stage in “exposure bias”.

---

> ### Author Response · Authors · 2018-01-05
> **Response to Reviewer1 - part 2**
>
>
> P5
> Why is this a generalized version of SeqGAN? The claim the discriminator value D(x_1..t | y) is the same as what you would get from a full-sequence generator using MCMC seems like a stretch.
>
> Ans: Thanks for your indication. We can understand our previous description is not appropriate, therefore we have rewrote the paragraph. StepGAN-Seq (the new name for the old ESGAN) is a generalized version of basic-SeqGAN, which uses 1-sample estimation instead of MCMC, because we can directly derive from the formulation. As you mentioned, we cannot directly say that StepGAN is a generalized version of SeqGAN using MCMC, and what we wanted to say is that StepGAN aims to approximate MCMC and to replace it.
>
> Did you not use a baseline?
>
> Ans: Yes, we do use baseline. In synthetic experiments, we estimated the baseline using the average of batch. In dialogue generation, we trained another value network to estimate the baseline.
>
> You might want to build in a little more motivation for these different update rules. I think I understand that (10) is meant to accumulate credit across the rest of the sequence, while (11) does not, but it would be good to have this clearly stated. Why do you think one would work better than the other?
>
> Ans: Thanks for your suggestion. We try to clarify this point in our updated version as (8) (the old (10)) viewing discriminator’s scores as rewards and (9) (the old (11)) viewing discriminator’s scores as expected returns (or Q value for state-action pair). We are not sure which one is better, but in our synthetic experiments, (9) has apparently higher accuracy than (10) in  “Counting”, while they do not show large difference in the other two tasks. Therefore we choose to use (9) in dialogue generation.
>
> P6
> “The generator G, in the mean time, struggle to maximize the likelihood of discriminator D” I don’t understand what this means.
>
> Ans: Thanks, we have rewrote that.
>
> What was the motivation for using the same model here? Is the energy in this formulation related in any ways to EBGAN? Could you do something similar with separate parameters? Why would be or why would this not be a good idea?
>
> Ans: Due to the performance of EBStepGAN (the new name of the old EBESGAN) does not show apparently difference with StepGAN and we have reached the page limitation, we have moved EBStepGAN to Appendix C and rewrote the description.
> The motivation of EBStepGAN is that we can use the same pretrained model for both generator and discriminator with only a variation of objective function. We indeed think that even though the generator and discriminator do similar things, it can still work.
> The spirit of EBGAN is viewing discriminator as an energy function, and we use a cross-entropy as the enery function here.
>
> I do like these synthetic tasks. I think that more analyses would be helpful in understanding what the model (and what competing models) are doing.
>
> Ans: Thanks ! We have add more analyses based on the real-world task: dialogue generation.
>
> P7:
> What is VLGAN in the table?
> Perhaps it would be worth exploring changing alpha through optimization?
>
> Ans: Yes, thanks for your realization. We have corrected the “VLGAN” -> “StepGAN” and “VLGAN-VRL” -> “StepGAN-Seq”. The table listed our exploring of alpha^G and the detail is described in Section 5.1.
>
> P8:
> It seems like many of the improvements in the table are marginal (with some exceptions): is it possible that ESGAN was optimized better?
> “auxiliary tricks” I would avoid this wording.
>
> Ans: Sorry for that we don’t have many time to conduct more experiments. Since we have done grid search over most hyper parameters, we thinks the most possible optimization way are changing the pretrained discriminator and adding value network for synthetic tasks.
>
> Other comments on experiments:
> It seems like the actual NLP part of this paper is quite sparse. Why was MaliGAN left out of the real experiments?
>
> Ans: Thanks for your advice. We have added MaliGAN in our new human evaluation experiments, which is listed in Table 3.

---

> ### Author Response · Authors · 2018-01-16
> **Response to Reviewer 1 - Updated Comments**
>
> Thank you very much for reading the responses and updating the comments.
>
> - I read the revision. The writing has improved, and the new experiments are good. That said, it is my opinion that the paper is still not quite ready for publication, mostly because the story behind the model doesn't lead to the conclusions being made from the experiments in a clean and consistent way. It's a bit like patchwork at this point. The authors I think will need to put some time into a rewrite, but the content itself is worth pushing forward.
> Some notes: "StepGAN a general version of SeqGAN, and can simulate the process of Monte-Carlo search with low extra computational cost:" This really is a strong claim that's not proven in any way in the paper.
>
> Ans: It is widely known that stepwise evaluation is critical for sequence generation by GAN, and Monte-Carlo search and REGS are previous attempt that we know. REGS does not perform as good as Monte-Carlo search in its original paper, but Monte-Carlo search has additional computational cost. We find that StepGAN behaves more like Monte-Carlo search without additional cost. We have observed that the variation of stepwise scores given by discriminator dominates the learning process of generator, which directly affects the final training results. We plot the variation (Figure 3), and can observe that StepGAN approximates Monte-Carlo search (MC-SeqGAN in Figure 3) better than REGS. Therefore, we claim that StepGAN can approximate Monte-Carlo. It is intuitive that StepGAN does not add any complexity to the original GAN.
> Thank you for your encouragement. We will further improve the paper to make it better if we further have chance to modify the paper.
>
> - "In typical reinforcement learning, the agent obtains a reward...": in typical RL settings, it's just as likely to find single or episodal rewards and this setting isn't limited to those where you have an extrinsic reward at each step.
>
> Ans: We think in an MDP, the reward is given when transition happens from any state s to another state s’. This definition does not exclude situations when receiving single or episodal rewards. These cases can be considered as the extrinsic rewards at every intermediate steps are zero and only the terminal step get a non-zero reward.
>
>
> - Why do you still have WGAN-GP when there are no accompanying numbers?
>
> Ans: The synthetic experiments (in Table 2) do include the results of WGAN-GP because WGAN-GP is a notable method, and we want to show that we have tested it. However, we have observed that WGAN-GP cannot improve the baseline under our settings. We therefore marked the numbers as dash (-), which you could interpret as worse than MLE. I will put the number of WGAN-GP in Table 2 if we have chance to modify the paper in the future.

---

### Official Review · AnonReviewer2 · 2017-11-27
**A well written paper with somehow weak experimental results**

**Rating:** 5
**Confidence:** 2

**Review:**

The authors present a new scheme for applying adversarial networks to dialog generation. The idea of why using adversarial networks is important in dialog generation is really well motivated in the paper and related works are discussed in details.

In the proposed approach, a more flexible discrimination score is obtained by treating independently each sub-sequence of the input. Technically speaking, the authors' contribution is to add a set of free parameters in the sub-sequence discriminator sum of equation 8. From a more general point of view, what is the key output of the paper, except to confirm that curriculum learning can help in dialog generation?

The experiments do not seem to show that a net performance improvement can be associated with the introduced free weights and what is a good strategy to tune them in an optimal way. In general, as the authors report also in the abstract, the performance of the proposed algorithm is 'comparable' with the state of the art but never outperforms other existing methods in a consistent way.

In fact, the performance of the algorithms depends strongly on the specific grammar used to generate the dataset and on the specific evaluation score. The human evaluation experiment is interesting but the proposed method is only compared with one other algorithm (seqGAN) and only two examples of the output are given explicitly.

The increase in computational cost due to the weighted sub-sequence evaluation is also poorly discussed.

Few more questions:
-through the experiments section, the authors focus on evaluating the set of possible good answers (softmax and coverage score) instead of the best answer (argmax). Why is this important for dialog generation? In the generation of a real conversation, shouldn t one always choose the argmax option? What would be a practical use of the second and third-best options?
-why softmax is always lower than argmax in the synthetic experiment and always higher than argmax in the human evaluation experiment?
-why MLE, which is used as initialization, does better than all optimized models in the first simulation? Why is the GAN approach expected to increase the coverage compared to MLE? And why, in general, this is not always the case?
-would it be possible to compare the output of the proposed methods with the output of a non-GAN conditional sequence generator (if any) on human-scored dialog?

---

> ### Author Response · Authors · 2018-01-05
> **Response to Reviewer2**
>
> In the proposed approach, a more flexible discrimination score is obtained by treating independently each sub-sequence of the input. Technically speaking, the authors' contribution is to add a set of free parameters in the sub-sequence discriminator sum of equation 8. From a more general point of view, what is the key output of the paper, except to confirm that curriculum learning can help in dialog generation?
>
> Ans: The proposed approach aims to approximate the expected return obtained from Monte Carlo search with a apparently lower computational cost. We have described it more clearly in Section 1: Introduction.
>
> The experiments do not seem to show that a net performance improvement can be associated with the introduced free weights and what is a good strategy to tune them in an optimal way. In general, as the authors report also in the abstract, the performance of the proposed algorithm is 'comparable' with the state of the art but never outperforms other existing methods in a consistent way.
>
> Ans: We think the most inconsistent experiment is “sequence” artificial grammar. The reason is very possible to be a very strong pretrained generator (accuracy~97%) for pre-trained discriminator. The discriminator is easily not pre-trained well with a very plausible generator. Therefore, the adversarial training cannot get advantages here. We have added this reason in Section 5.1.
>
> In fact, the performance of the algorithms depends strongly on the specific grammar used to generate the dataset and on the specific evaluation score. The human evaluation experiment is interesting but the proposed method is only compared with one other algorithm (seqGAN) and only two examples of the output are given explicitly.
>
> Ans: Thanks for your advice. We add MaliGAN, REGS, beam search, and MMI for comparison. Therefore we conduct a new human evaluation process and renew the table. For more examples, we have put them in Appendix D.
>
> The increase in computational cost due to the weighted sub-sequence evaluation is also poorly discussed.
>
> Ans: We add one sentence in Section 4 for StepGAN (the new name for the old ESGAN) and one sentence in Section 3.3.1 for Monte Carlo search. We hope you can find it useful.
>
> Few more questions:
> -through the experiments section, the authors focus on evaluating the set of possible good answers (softmax and coverage score) instead of the best answer (argmax). Why is this important for dialog generation? In the generation of a real conversation, shouldn t one always choose the argmax option? What would be a practical use of the second and third-best options?
>
> Ans: We think the practical use of options other than the argmax one is that we can further transfer the style of the generation. Because learning the set of possible good answers means the model learns the underlying distribution, this enable the model to generate assigned style from the set of possible answers.
>
> -why softmax is always lower than argmax in the synthetic experiment and always higher than argmax in the human evaluation experiment?
>
> Ans: In synthetic experiments, the accuracy only considers whether the response is correct. Therefore, if the argmax has learned pretty good, the softmax would only has lower accuracy because it diverse from the correct answer.
> However, in dialogue generation, the human score was evaluated over whether the response was good, which included both coherence and the information provided by the response. Since the softmax results were much diverse and resulted in more interesting response, we thought this diversity would also be considered by human critics.
> Besides, due to the high variety of softmax and the high costs of inviting human critics, we do not use softmax in our new human evaluation experiment. Also, to better analyze where’s the improvement, we separate the coherence and the information provided into two scores for human evaluation.
>
> -why MLE, which is used as initialization, does better than all optimized models in the first simulation? Why is the GAN approach expected to increase the coverage compared to MLE? And why, in general, this is not always the case?
>
> Ans: We have added explanation in Section 5.1. MLE in the first simulation has a very high accuracy, which then reduces the performance of pre-trained discriminator. The GAN-based algorithms then cannot get advantages based on this poor pre-trained discriminator. This is why we thought the first simulation is different from others.
>
> -would it be possible to compare the output of the proposed methods with the output of a non-GAN conditional sequence generator (if any) on human-scored dialog?
>
> Ans: Thanks for your suggestion. We add beam search and MMI in our new human-scored dialog experiments. The result is listed in Table 3.

---

### Decision · Program_Chairs · 2018-01-29
**ICLR 2018 Conference Acceptance Decision**

**Decision:**

Reject

**Comment:**

Using generative adversarial networks for conditional sequence generation with improved computational cost is a promising approach, unfortunately this work falls short of convincing that it would be more useful than existing methods.